# How Carbon Nanoparticles, Arbuscular Mycorrhiza, and Compost Mitigate Drought Stress in Maize Plant: A Growth and Biochemical Study

**DOI:** 10.3390/plants11233324

**Published:** 2022-12-01

**Authors:** Emad A. Alsherif, Omar Almaghrabi, Ahmed M. Elazzazy, Mohamed Abdel-Mawgoud, Gerrit T. S. Beemster, Renato Lustosa Sobrinho, Hamada AbdElgawad

**Affiliations:** 1Department of Biology, College of Science and Arts at Khulis, University of Jeddah, Jeddah 23218, Saudi Arabia; 2Department of Botany and Microbiology, Faculty of Science, Beni-Suef University, Beni Suef 2722165, Egypt; 3Department of Biology, College of Science, University of Jeddah, Jeddah 23218, Saudi Arabia; 4Chemistry of Natural and Microbial Products Department, Pharmaceutical and Drug Industries Research Division National Research Centre, Dokki, Giza 12622, Egypt; 5National Natural Products Research Center, College of Pharmacy, University of Mississippi, Oxford, MS 38655, USA; 6Department of Medicinal Plants and Natural Products, Desert Research Center, Cairo 11753, Egypt; 7Integrated Molecular Plant Physiology Research, Department of Biology, University of Antwerp, 2000 Antwerp, Belgium; 8Department of Agronomy, Federal University of Technology—Paraná (UTFPR), Pato Branco 85503-390, PR, Brazil

**Keywords:** proline, photosynthesis, corn, fatty acid, organic acid, sustainable agriculture

## Abstract

Drought negatively affects crop growth and development, so it is crucial to develop practical ways to reduce these consequences of water scarcity. The effect of the interactive potential of compost (Comp), mycorrhizal fungi (AMF), and carbon nanoparticles (CNPS) on plant growth, photosynthesis rate, primary metabolism, and secondary metabolism was studied as a novel approach to mitigating drought stress in maize plants. Drought stress significantly reduced maize growth and photosynthesis and altered metabolism. Here, the combined treatments Com-AMF or Com-AMF-CNPs mitigated drought-induced reductions in fresh and dry weights. The treatments with AMF or CNPS significantly increased photosynthesis (by 10%) in comparison to the control plants. Results show that soluble sugars were accumulated to maintain the osmotic status of the maize plant under drought stress. The level and metabolism of sucrose, an osmo-protectant, were increased in plants treated with Com (by 30%), which was further increased under the triple effect of Com-AMF-CNPs (40%), compared to untreated plants. This was inconsistent with increased sucrose-phosphate synthase and sucrose-P-synthase activity. The combined treatment Com-AMF-CNPs increased the levels of oxalic and succinic acids (by 100%) and has been reflected in the enhanced levels of amino acids such as the antioxidant and omso-protectant proline. Higher increases in fatty acids by treatment with CNPS were also recorded. Com-AMF-CNPs enhanced many of the detected fatty acids such as myristic, palmitic, arachidic, docosanoic, and pentacosanoic (110%, 30%, 100%, and 130%, respectively), compared to untreated plants. At the secondary metabolism level, sugar and amino acids provide a route for polyamine biosynthesis, where Com-AMF-CNPs increased spermine and spermidine synthases, ornithine decarboxylase, and adenosyl methionine decarboxylase in treated maize. Overall, our research revealed for the first time how Cmo, AMF, and/or CNPS alleviated drought stress in maize plants.

## 1. Introduction

The production of food crops around the world is most negatively impacted by severe drought. Due to the limited soil water content, drought stress limits plant growth and development, including leaf wilting, a decrease in plant height, and biomass accumulation during its early vegetative stage [1,2]. In this regard, plants cannot absorb nutrients, which results in reduced plant growth [3]. Saudi Arabia is one of the most vulnerable countries to climate change due to its continental climate, harsh winters, hot summers, and minimal rainfall [4]. It is characterized by large natural desert landscapes with limited water and nutritional resources, posing a challenge to the kingdom’s food security concerns [5]. In response to water stress, plants must adapt morphologically, physiologically, and biochemically to combat the loss of water and maintain their hydric condition [6]. Accumulation of osmolytes, including polyols, free amino acids, and soluble sugars, is a well-known plant response to water deficit [7,8]. Osmolytes are compatible solutes that adjust cellular osmotic potential, protect membranes and proteins, and protect against oxidative damage by scavenging ROS [9,10].

Plant waste composts (Comp) are being introduced as effective fertilizers to improve plant growth and yield in the face of environmental change [11]. In organic farming, using leftover vegetables may increase soil fertility, which will enhance plant growth and nutritional value. For example, Bokobana et al. [12] reported that the Comp treatment enhanced controllable maize growth and drought tolerance. Increasing the plant’s access to nutrients by using Comp can be further improved by adding plant growth-promoting microorganisms such as arbuscular mycorrhizal fungus (AMF). In this regard, organic matter can be decomposed and recycled by AMF, which plays a key role in the soil environment by channeling nutrients from the soil to their host plant and binding soil particles together into aggregates, forming and maintaining the pore spaces [13]. Most of the world’s terrestrial ecosystems are supported by AMF, whereas symbiotic relationships could be developed between plant roots and certain species of soil fungi [14,15]. As a result, AMF considers it to be one of the most effective methods of increasing plant stress tolerance under environmental stress conditions [16].

Like the protective effect of AMF, nanomaterials can be used to boost crop productivity by controlling the infusion of nutrients and improving agricultural output efficiency. For instance, Khadakovskaya et al. [17] presented carbon nanotubes as a nano-fertilizer, reporting that numerous crops experienced growth increases [18,19]. Additionally, in environmental and agricultural applications, herbicide mobility may be decreased by nanoparticles with significant adsorption capabilities, such as carbon nanoparticles (CNPs). CNPs enhance plant photosynthesis, crop growth, and water intake, according to Mukherjee et al. [20]. Moreover, it was reported that CNPs improved mineral consumption efficiency [21]. It is also known to improve tissue chemical composition and tissue quality (REF) [22].

Saudi Arabia uses corn, a summer grain with a significant economic value, for human nourishment. The government is reliant on imports because domestic production cannot meet all the country’s consumption needs for maize [23]. In the seedling stage, maize crops are particularly vulnerable to drought, causing a severe reduction in yield. The goal of this study is to find out how the interaction between Com, AMF, and CNPs can be used as a new way to help maize plants tolerate drought stress. We test the hypotheses that (1) drought stress reduces maize plant growth by altering their physiology and primary and secondary metabolism; (2) this drought stress effect is reduced by individual treatment of Comp, AMF, or CNPs; and (3) the effect of Comp, AMF, or CNPs is further strengthened by their combination. The current study sought to better understand the potential contribution of CNPs to improving the interactive effect of AMF and Comp on corn crops during drought. The development of suitable strategies or treatments to raise plants’ tolerance for drought stress may benefit from this information.

## 2. Material and Methods

### 2.1. Compost Formation

The raw leafy vegetable waste used for composting was collected from the Jeddah wholesale vegetable market in Jeddah city. The vegetable waste was chopped into small pieces (approximately 1 cm particle size), and raw material was introduced to three digester cells during the preprocessing stage of composting (3 m long, 1 m wide, and 1 m high; non-covered cement containers). To maximize microbial activity, the moisture content of these raw materials was modified to be between 55 and 65 %, and urea was used to change the initial C/N ratio of the composting materials to between 5 and 6. To hasten the composting process, a blend of *Phanerochaete chrysosporium* Burdsall (40% by volume) and *Trichoderma* spp. (60% by volume) was introduced to the composting materials. Water was measured weekly during the composting process to keep it between 60 and 70%. On the third day, the pile’s temperature rose to 50–60 °C and remained there for almost six days. The composting process was completed after 42 days when the temperature of the pile reached the surrounding air. The finished product was sent to the testing location for use. The obtained compost had the following physiochemical properties: pH is 6.88, total nitrogen is 2.13 percent, C/N is 8.7, EC is 0.92 ms/cm, total phosphorus is 0.25 percent, and bulk density is 0.36, as an indicator of composite homogeneity. Lathyrus sativus L.’s strength was similarly decreased in conditions of water shortage.

### 2.2. Carbon Nanoparticles (CNPs)

Water-dispersible CNPs were purchased from Vulpes Inc. (St. Louis, MO, USA). They were composed of C (63%), O (34%), H (1.6%), and N (1.4%). The CNPs were 90–120 nm in wavelength and had an average zeta potential of 66.9 mV (Malvern Nano-ZS Zetasizer). The specific surface area of 35–50 m^2^g^−1^ and porosity ranges of tested CNPs (7–11%) were assayed.

### 2.3. Experimental Set-Up

Sowing was started in May, and the experiment continued for two months until flowering. Descriptions of climatic conditions and properties of the experimental soil are provided in Appendix A. The soil at the experimental site had a sandy loam texture, and the research field had been carefully prepared. A few of the test soil’s physicochemical properties are shown in Table 1. Each plot had ridges and hills that were separated by 70 and 25 cm, respectively, and measured 10.5 m^2^ (3 m × 3.5 m). One of the stores authorized by the Saudi Ministry of Agriculture to sell crop seeds provided maize seeds (variety Giza 81). For eight hours, maize seeds were allowed to prime in CNP concentrations (80 μg/mL). The priming techniques speed up the CNPs’ quick penetration into maize seeds and reduce their direct contact with soil bacteria by ignoring the chance of CNP seeping out into the soil. On one side of the ridge, control and primed seeds were manually sown as three grains per hill. In the experiment, it was taken into consideration that the control plants are at a large distance from the treated plants. Moreover, a plastic barrier was placed inside the soil (1 m deep) to separate the control plants from the treated plants. The treatments were set up in a strip-plot design with four replications, with the vertical plots being the compost treatments (control, 100% compost; AMF; AMF + 50% compost; AMF + B. amyloliquefaciens + 50% NPK). Soils were infected with a pure commercial inoculum of *Rhizophagus irregularis* to test for AMF (MUCL 41,833 purchased from GINCO). The irrigation treatments produced horizontal plots (well-watered condition, WW, and withholding irrigation at R5, DS). In well-watered conditions, irrigation was given every 15 days throughout the full growing season. Irrigation was turned off during seed development for two weeks to generate a condition of drought stress (14 days from sowing). While the soil naturally dried during the maturation process, soils at the seedling stage were given 25% of the entire water requirement (5 percent). Other corn-growing practices in Saudi Arabia were to follow those followed by local farmers in the area. The used maize variety has simplified crop husbandry, due to more effective control of pests and weeds. To prevent weeds from competing and interfering with different treatments, weeds were mechanically and chemically (treatment with atrazine 4 L/ha) removed. Pests were also controlled by adding chlorpyrifos (0.8 L/ha). N fertilization as urea (140 kg N/ha) was incorporated into the soil.

### 2.4. Environmental Growth Condition

The Khulais region, located in the Arabian Shield at 22°8’55” N and 39°20’52” E, underwent the experimental climate condition in 2021. High summer temperatures and mild winters define the climate (Appendix A). This area receives 35.1 mm of precipitation on average per year. Precipitation, which ranges from 0 to 70 mm, is infrequent, unpredictable, and scarce. The monthly peak and low air temperatures are 37.4 degrees in June and 13.4 degrees, respectively.

### 2.5. Soil Structure and Composition

Chemical characteristics of the study’s soils are presented in Appendix A. In comparison to the control, soils that had compost, AM, or both (compost-AM) added had higher nutritional values. In terms of most cations, anions, accessible nutrients, and micronutrients, the soil to which the mixture (Comp-AM) was added recorded the greatest values, followed by the soil to which only compost was added. The highest Fe and Zn levels were produced when only AM was given to the soil. The amount of clay, organic matter, and salinity rose while compost, AM, and their mixture were added.

### 2.6. Determination of Photosynthesis Rate

A portable photosynthesis device was used to determine the light-saturated photosynthetic rate (LI-6400; LI-COR). In the leaf chamber, the CO_2_ concentration was fixed at 400 μmol mol^−1^, while the temperature was determined at 25 °C.

### 2.7. Determination of Rubisco Activity

Rubisco activity was analyzed according to Sulpice et al., 2007. It was expressed as the conversion rate of glycerate kinase (3-PGA) of extracted leaf samples (µmol 3-PGA m^−2^ min^−1^). The activity of rubisco was determined without incubation of the extract in the presence of 10 mM HCO^3−^ and 20 mM Mg^2+^ to convert the non-carbamylated rubisco into the carbamylated form.

### 2.8. Determination of Sugars

Sugar extraction was performed by using 50 mM Tris-acetate-EDTA (TAE) (pH 7.5) containing a mixture of polyclar (0.15%), Na azide (0.02%), PMSF (2 mM), mercapto-ethanol (1 mM), mannitol (10 mM), and NaHSO_3_ (12 mM). The mixture was centrifuged (15,000× *g*, 4 °C, 10 min). Afterward, 0.2 mL of the mixture was taken and heated at 90 °C for 5 min; then, it was allowed to cool down. This part was taken and moved to a mixed-bed Dowex column (300 mL Dowex H^+^, 300 mL Dowex Ac^−^; both 100–200 mesh; Acros Organics, Morris Plains, NJ, USA). Thereafter, elution was performed six times with 0.2 mL ddH_2_O, and quantification of soluble sugars was performed by using (HPAEC-PAD) [24]. Analysis and detection were performed at 30 °C. A total of 25 μL of sample was injected into a Guard CarboPac PA 100 (2 × 50 mm) in series with an analytical CarboPac PA 100 (2 × 250 mm). The flow rate was 200 μL per min. Sugars were eluted in 90 mM NaOH, with an increasing Na-acetate gradient: from 0 to 6 min, the Na-acetate concentration increased linearly from 0 to 11 mm; from 6 to 18 min, from 10 to 110 mm; and from 16 to 26 min, from 100 to 175 mm. The columns were then regenerated with 500 mM Na-acetate for 1 min and equilibrated with 0.09 M NaOH for 11 min for the next run. Data were recorded and processed with Chromeleon software.

For determination of sugar-related enzymes, the non-heated supernatant was used according to [25]. The determination of invertase activity was performed in TAE buffer and sucrose (100 mM). The mixture was incubated at 30 °C, and then the reaction was stopped by keeping an aliquot for 5 min at 90 °C. Aliquots (0.2 mL) were incubated (at 30 °C) with 100 μL reaction mixture containing 100 mM sucrose in TAE buffer pH 8.5 (neutral invertase) and 0.02% (w/v) Na-azide. Reactions were stopped by keeping an aliquot for 5 min in a water bath at 90 °C. Likewise, the concentrations of glucose and fructose were evaluated using HPAEC-PAD (Dionex, Sunnyvale, CA, USA). On the other hand, the determination of sucrose phosphate synthase (SPS) was performed by using HEPES buffer (1 mL, pH 8) containing fructose-6-phosphate and UDP-glucose. The reaction was processed at 37 °C for 20 min, and then NaOH (30%) was added to stop the reaction. Evaluation of starch synthase activity was performed in a mixture containing citrate and glycogen (Nishi et al., 2001). Meanwhile, amylase activity was detected in a starch solution containing I_2_/KI (0.05%) in HCl (0.05% as well), and then the reading was taken at 620 nm [25].

### 2.9. Determination of Organic Acids

Organic acids were detected in the tested plants by using HPLC under the following conditions: 0.001 N sulfuric acid, at 210 nm, 0.6 mL min^1^. The HPLC system consisted of a liquid chromatographer (Dionex, Sunnyvale, CA, USA) and a detector (LED, ultimate 3000), in addition to a pump (LPG-3400A), a column thermostat (TCC-3000SD), and an autosampler (EWPS-3000SI). Separation of organic acids was conducted through an Aminex HPH-87 H (300 × 7.8 mm) column coupled with IG Cation H (30 × 4.6) precolumn of the Com-Red firm (at 65 °C). Data analysis and interpretation were performed using chromeleon v.6.8 computer software [21].

### 2.10. Determination of Amino Acid Levels and Metabolism

Amino acids were analyzed according to [26,27]. Extraction was performed by using 100 mg of plant samples in 5 mL of 80% ethanol, and then centrifugation was conducted (14,000× *g*, 25 min). Afterward, the supernatant was taken and resuspended in chloroform (5 mL), while the residue was extracted with H_2_O (1 mL). The supernatant and pellet were resuspended in chloroform and centrifuged (8000× *g*, 10 min). Detection and quantification of amino acids were performed by using UPLC (Waters Acquity, TQD). The chromatograph was connected online to a triple-quadrupole mass spectrometer detector (TQD) with an electrospray ionization (ESI) interface. The UPLC was supplied as BEH amide column. The elution was performed by using the gradient system (A: ammonium formate (84%), formic acid (6%), and acetonitrile (10%), and B: acetonitrile and formic acid (2%)). MassLynx software version 4.1 from Waters (Milford, MA, USA) was used to control the instruments and collect and analyze the data.

### 2.11. Determination of Fatty Acids

Fatty acids were detected in treated and non-treated plants by using GC/MS (Hewlett Packard, Palo Alto, CA, USA). The analysis was performed using an Agilent single-quadrupole mass spectrometer with an inert mass selective detector (MSD-5975C detector, Agilent Technologies, Santa Clara, CA, USA) coupled directly to an Agilent 7890A gas chromatograph which was equipped with a split–splitless injector, a quick-swap assembly, an Agilent model 7693 autosampler, and an HP-5MS fused silica capillary column (5% phenyl 95% dimethylpolysiloxane, 30 m × 0.25 mm i.d., film thickness 0.25 μm, Agilent Technologies, Santa Clara, CA, USA). The temperature of the oven was held at 80 °C for 2 min, raised to 200 °C at 5 °C/min (1 min hold), and then to 280 °C at 20 °C/min (3 min hold). A 1.0 uL sample was injected using a split mode (split ratio, 1:10). Data analysis and interpretation were performed using NIST 05 and Golm Metabolome (http://gmd.mpimp-golm.mpg.de (accessed on 5 May 2022) [28].

### 2.12. Polyamine Metabolism

Plant samples were extracted in cold perchloric acid [29]. After centrifugation for 30 min at 14,000× *g*, we applied dansyl chloride to derivatize the sample supernatants and standards to detect free polyamines. After hydrolyzation at 110 °C overnight, 6 N HCl was added for the formation of conjugated polyamines. The reverse-phase HPLC (Shimadzu SIL10-ADvp; C18 column) was used to measure the concentrations of dansylated polyamine derivatives. For measuring the activities of polyamine-related enzymes, plant shoots and roots were homogenized in 100 mM KPO_4_, pH 7.5, and they were centrifuged for 20 min at 25,000 RPM. Ornithine decarboxylase (ODC) and arginine decarboxylase (ADC) enzyme activity was measured by measuring the labeled CO_2_ (Birecka et al., 1985), which was liberated from L-[l-14C] Ar (55 mCi mmol^−1^) and L-[l-14C] Orn (55 mCi mmol^−1^). The trapped radioactivity was measured with a liquid scintillation counter. The activity of spermidine (Spd) synthase was assayed in a reaction mixture of pH 8.0 Tris-HCl (0.1 M). The production of 5′-deoxy-5′-methylthioadenosyne was quantified by using a fluorescence detection method (a reverse-phase HPLC equipped with a fluorescence detector). Spermine synthase (SpmS) activity was measured by the production of methylthioadenosine [30].

### 2.13. Statistical Analysis

All results were expressed as the means of five biological replicates (*n* = 5). Statistical analysis was performed using one-way ANOVA in the SPSS 22 (Tukey test, *p* ≤ 0.05). Data normality was checked by using Levene’s test. Meanwhile, the hierarchical cluster analysis Euclidean distance was performed by using the R stat software package (version 4.5.0, the R).

## 3. Results

### 3.1. Effect of Different Treatments on Maize Growth

The ability of Com, AMF, and/or CNP_S_ to enhance plant growth and biomass production was investigated in our study. When maize plants were treated with CNPs, there were slight increases in biomass accumulation (expressed as fresh and dry weights; FW and DW), while no changes were observed under the sole treatment with Com or AMF in comparison to the control plants (Figure 1). Under the double effect of CNP_S_ and AMF- CNPs, there were also slight increases in biomass production; however, the triple effect of Com-AMF-CNP_S_ did not induce significant changes in FW but increased the DW (by 40%), when compared with the control plants (Figure 1). On the other hand, when drought conditions were applied to maize plants, the treatment with Com enhanced both FW and DW (by 30% and 50%, respectively), which were further increased when plants were treated with CNPs. Interestingly, the combined treatment with Com-AMF had a more enhancing effect on FW and DW (by 70–80%), which were also further enhanced by treatment with Com-AMF-CNPs, as compared with the sole drought treatment. This could clearly indicate the ability of combined treatments with Com-AMF or Com-AMF-CNPs to recover the reduction in FW and DW caused by drought stress (Figure 1).

### 3.2. Photosynthetic Parameters

The photosynthetic activity of plants is known to be induced by mycorrhizal fungi as well as compost. In this regard, our results show that photosynthesis was almost not affected by compost treatment; however, the treatments with AMF or CNPs caused significant increases in photosynthesis (by 10%) in comparison to the control plants (Figure 2a). Such increases were much more enhanced when AMF was combined with CNPs or under the triple effect of Com-AMF-CNPs (30%) as compared with the control plants. Under drought conditions, a gradual increase in photosynthesis could be triggered under all single and combined treatments, whereas the treatments with Com or AMF increased the photosynthetic activity (20%), which was further increased under treatment with CNPs. The interaction between compost and mycorrhiza, as well as the Com-AMF-CNPs, resulted in higher increments in photosynthesis (by 50% and 60%, respectively), as compared with the sole drought treatment (Figure 2a).

To understand the induced effects of Com, AMF, or CNPs on the photosynthetic efficiency, we further measured the activity of ribulose-1,5-bisphosphate carboxylase-oxygenase (RuBisCO), which is the enzyme involved in carbon fixation to produce sugars. The results demonstrate that no significant changes in RuBisCO activity were reported under all the single treatments with Com, AMF, or CNPs. When compost was combined with AMF, there was a significant increase in RuBisCO activity and photosynthetic (by 30%) as compared with the control plants (Figure 2a,b). A similar effect was also obtained under a combination of AMF and CNPs, or Com-AMF-CNPs. The drought conditions also stimulated a significant increase in RuBisCO when plants were treated with compost (by 30%), which was further enhanced by CNPs (by 50%) as compared with the sole drought treatment. Figure 2a shows that the interactive impacts of Com or AMF, as well as the Com-AMF-CNPs, were reflected in higher increases in RuBisCO activity (by 50% and 70%, respectively). This could also show that the combined treatments worked to bring back the photosynthetic efficiency that was lost when the plant was stressed by drought.

### 3.3. Sugar Metabolism

Sugars could play an effective role in plants, particularly under stressful conditions, where they might function as osmoregulators (e.g., sucrose). Moreover, sugars may be involved in the biosynthesis of antioxidant metabolites, which in turn could play a role in the plant’s mechanism against stress factors. Thus, we analyzed the changes in individual and total soluble sugars of maize plants treated with Com, AMF, or CNPs under control or drought conditions. Regarding the individual sugars, all the single and combined treatments did not induce significant changes in the amounts of glucose and fructose, while sucrose was only enhanced under a combination of Com and AMF (by 20%) or Com and CNPs (50%) when compared with the control plants (Table 1). Under drought conditions, the glucose level was not changed by any of the treatments, while fructose was increased only by treatment with Com or the triple combination of Com-AMF-CNPs (by 90%). Significant increases in sucrose were also observed when plants were treated with Com (by 30%), which were further increased under the triple effect of Com-AMF-CNPs (by 40%), on drought-stressed plants.

To get more insight into the changes in sugar contents under the impact of Com, AMF, and/or CNPs, we also determined the total soluble sugars in maize plants, which were remarkably increased by 30% due to treatment with AMF or CNPs as compared with the control values. Among the combined treatments, the treatment with Com and CNPs was the only one that induced a 30% increase in the total soluble sugars. The drought conditions enhanced the total soluble sugars when using the AMF treatment (30%), which was further increased under all treatments, particularly by the combined effect of Com-AMF-CNPs.

To investigate the mechanism underlying the changes in sugar metabolism, we investigated the sucrose biosynthetic enzyme sucrose-phosphate synthase (sucrose-P-synthase), which is involved in sucrose synthesis (Table 1). The treatments with only Com or CNPs did not increase the activity of sucrose-P-synthase. However, when compared to the control plants, the treatment with only AMF did increase the activity of sucrose-P-synthase by 50%. In addition, data in Table 1 show that the sucrose-P-synthase was remarkably activated because of most of the combined treatments, particularly when Com was combined with CNPs (90%) as compared with the control plants. Under drought stress, the sucrose-P-synthase was not affected by any of the treatments, except for the combined treatment with Com and AMF (80%).

Further, we evaluated the changes in sucrose biodegradation enzymes, i.e., invertase, which is incorporated in the degradation of sucrose into glucose and fructose. Under control conditions, no changes were observed in invertase activity under all single treatments. When maize plants were treated with a combination of Com-AMF or Com-AMF-CNPs, there were significant increases in the activity of invertase (by 50% and 90%, respectively) as compared with the control values (Table 1). Meanwhile, under drought conditions, the sole treatment with Com or AMF enhanced the invertase activity (30%), while no changes were observed under treatment with CNPs. The interaction between Com and AMF led to higher increments in invertase activity (90%) when compared with the sole treatment with drought (Table 1). It was discovered that sucrose levels correlated with the rate of biosynthetic and biodegradation enzymes, especially when Com-AMF-CNPs were combined.

### 3.4. Organic Acid Level

The changes in the number of sugars under different treatments might induce enhancing effects on the tricarboxylic acid cycle intermediates, such as organic acids. In this regard, we measured the changes in the levels of organic acids in maize plants grown under treatment with Com-AMF-CNPs under control or drought conditions. Among the detected organic acids under control conditions, malic acid had the highest concentration, while fumaric acid had the lowest level. Under control conditions, the individual treatment with Com led to a dramatic increase in citric acid (200%), as well as significant decreases in malic and isobutyric acids (30% and 60%, respectively), while oxalic, succinic, and fumaric were not affected (Table 2). Citric acid was the only organic acid that was enhanced by AMF treatment (by 100%) when compared with the control plants. The treatment with CNPs increased malic and citric acids (by 50% and 300%, respectively), but reduced oxalic and isobutyric acids, while succinic and fumaric acids were not changed. The combined treatments did not affect the levels of oxalic and fumaric acids. Meanwhile, a combination of Com-AMF reduced the amounts of succinic and isobutyric acids but increased the malic acid content (50%) as compared with the control plants. Moreover, the combined treatment with Com and CNPs increased the content of citric acid (by 110%) but decreased the level of succinic acid (by 50%). The combination of Com-AMF-CNPs did not affect all the detected organic acids under control conditions (Table 2).

On the other hand, under drought conditions, the treatment with Com caused significant increases in only oxalic and citric acids (70% and 120%, respectively), while the rest of the organic acids were not affected. AMF treatment enhanced oxalic acid (by 120%) but reduced the malic acid content (by 20%) when compared with the drought-stressed plants. When plants were treated with CNPs, there were significant reductions in malic, succinic, and citric acids, while the contents of oxalic, isobutyric, and fumaric acids were not changed. The combined treatment of Com-AMF-CNPs also positively affected the levels of oxalic and succinic acids (by 100%), while the rest of the organic acids were not affected (Table 2). So, it could be noted that the organic acids of maize plants interacted differently with treatments with Com-AMF-CNPs under control or drought conditions.

### 3.5. Amino Acid Metabolism

Under stress conditions, the increased levels of amino acids could play a role as osmoprotectants, which help the plant cope with the unfavorable conditions. Therefore, we investigated the changes in the individual amino acids of maize plants as affected by different treatments with Com-AMF-CNPs under control or drought conditions. Under control conditions, glycine had the highest concentration among the detected amino acids, while leucine had the lowest amount (Table 3). Under control conditions, the treatment with Com significantly enhanced the content of alanine, arginine, asparagine, threonine, valine, glutamic acid, and cysteine, while the AMF treatment increased the levels of all the detected amino acids, except for glycine and asparagine. Moreover, the treatment with CNPs promoted many of the detected amino acids, in comparison to the control values. When maize plants were grown under the combined treatment of Com or AMF, a few amino acids were significantly increased, e.g., lysine, alanine, arginine, glutamine, and cysteine. Moreover, a combination of Com and CNPs promoted the contents of glycine, histidine, alanine, arginine, leucine, threonine, valine, serine, glutamic acid, aspartate, and cysteine. When AMF was combined with CNPs, there were significant increments in alanine, arginine, glutamine, isoleucine, leucine threonine, valine, phenylalanine, glutamic acid, and cysteine. Furthermore, Table 3 shows that the triple effect of Com-AMF-CNPs was reflected in the enhanced levels of histidine, arginine, ornithine, glutamine, isoleucine, leucine, valine, phenylalanine, glutamic acid, and cysteine (by 40%, 100%, 100%, 210%, 100%, 220%, 80%, 20%, 800%, and 100%).

Under drought conditions, the treatment with Com did not induce the majority of detected amino acids, except for arginine, phenylalanine, glutamic acid, and aspartate. Similarly, under AMF treatment, no significant effects were observed on most detected amino acids, except for asparagine, isoleucine, leucine, and cysteine. Meanwhile, the treatment with CNPs resulted in significant increments in lysine, glutamine, asparagine, leucine, valine, glutamic acid, and cysteine, as compared with the drought-stressed plants (Table 3). The combined treatment with Com-AMF did not exert significant effects on most detected amino acids, except for glutamine, asparagine, phenylalanine, and glutamic acid. Positive responses were also obtained in the contents of lysine, glutamine, asparagine, leucine, valine, phenylalanine, glutamic acid, aspartate, and cystine when plants were treated with Com-AMF-CNPs. So, it could be noticed that the amino acids of maize plants interacted differently with treatment with Com-AMF-CNPs, particularly under their combination. Table 3 Effect of different treatments on amino acids (mg g^−1^ DW) of maize under control and drought (D) condition. Different letters indicate statistically significant difference between means for the different treatments at significance level at least *p* ≤ 0.05.

To understand the reasons underlying the increases in the levels of proline under different treatments, we analyzed the proline metabolic enzymes, i.e., pyrroline-5-carboxylate synthetase (P5CS), pyrroline-5-carboxylate reductase (P5CR), ornithine-d-aminotransferase (OAT), and ProLINE dehydrogenase (ProDH) (Table 4). Under control conditions, the treatment with Com, AMF, or CNPs did not change proline biosynthesis, while the Com and CNPs or AMF and CNPs treatments increased P5SC and P5CR, but this did not alter the proline level. On the other hand, drought stress increased proline biosynthesis enzymes (P5SC, P5CR, and OAT).

This increase in enzyme activity was further induced by Com, AMF, or CNPs. Moreover, a combination of Com and AMF or Com, AMF, or CNPs had the largest effects on proline metabolism (Table 4). Overall, the combined effect of Com, AMF, or CNPs was more beneficial than their individual treatments for increasing the levels of proline and its biosynthesis under drought stress.

### 3.6. Fatty Acid Level

Fatty acids are considered among the tricarboxylic acid cycle intermediates and hence could be affected by stress factors. Therefore, we evaluated the individual fatty acids (saturated and unsaturated) of maize plants treated with Com, AMF, and CNPs under controlled or drought conditions. This might help to define the plant’s nutritional value. Under controlled conditions, palmitic acid (C16:0) had the highest concentration among the saturated fatty acids, while oleic acid (C18:1) had the highest level among the unsaturated fatty acids (Table 5). Regarding the saturated fatty acids, all the individual and combined treatments did not induce significant effects on palmitic (C16:0), arachidic (C20:0), and pentacosanoic (C25:0) acids. The treatment with Com significantly increased the contents of myristic and stearic acids (20%) but did not affect the levels of heptadecanoic, docosanoic, and tricosanoic acids. Meanwhile, the treatment with AMF enhanced the levels of myristic, stearic, and docosanoic acids (20%), while heptadecanoic and tricosanoic acids were not affected, as compared with the control plants. When plants were treated individually with CNPs, significant increases were induced in myristic, stearic, and docosanoic acids. Similar effects were also observed under the combined treatments, whereas the treatment with Com-AMF increased the concentrations of stearic and docosanoic acids but did not induce significant changes in myristic, heptadecanoic, and tricosanoic acids when the treated plants were compared with their respective controls. When Com was combined with CNPs, significant increases in heptadecanoic, stearic, docosanoic, and tricosanoic acids were clearly observed. Table 5 shows that the combination of Com-AMF-CNPs induced remarkable increments in only heptadecanoic and tricosanoic acids (by 110%).

Saturated fatty acids reacted differently to Com-AMF-CNP treatments under drought conditions. Heptadecanoic and stearic acids were not changed under any of the individual or combined treatments. When maize plants were grown under the effect of Com, there were significant increases in the levels of some detected fatty acids, i.e., myristic, docosanoic, and tricosanoic acids (by 100%, 30%, and 10%, respectively). The AMF treatment also caused remarkable increments in myristic, arachidic, docosanoic, and tricosanoic acids (by 110%, 20%, 80%, and 20%, respectively). Higher increases were induced in arachidic, docosanoic, and tricosanoic acids by treatment with CNPs. A combination of Com-AMF-CNPs had the upper hand to enhance most of the detected fatty acids, i.e., myristic, palmitic, arachidic, docosanoic, and pentacosanoic (by 110%, 30%, 100%, and 130%, respectively), as compared with the sole drought treatment (Table 5).

On the other hand, the unsaturated fatty acids showed wide variability in response to the different treatments with Com, AMF, and/or CNPs. In addition, Table 5 shows that the sole treatments with Com or AMF did not affect most of the detected fatty acids, except for linoleic and linolenic acids, while the treatment with CNPs significantly increased the content of only linoleic acid and decreased eicosanoic acid, in comparison to the control plants. All the combined treatments did not exert significant effects on both eicosenoic and tetracosenoic acids. The treatment with a combination of Com-AMF or Com-CNPs did not enhance most of the detected fatty acids, except for linoleic, while the combined treatment with AMF-CNPs enhanced only heptadecanoic and linoleic acids (by 10% and 90%, respectively), as compared with the control plants. Meanwhile, the triple action of Com-AMF-CNPs increased the concentrations of palmitoleic, oleic, and linoleic acids. A similar scenario was developed under drought stress, where the sole treatments with Com or AMF did not affect most fatty acids, except for heptadecenoic and oleic acids, while the treatment with CNPs resulted in significant increases in heptadecenoic, oleic, linoleic, and linolenic acids (by 100%, 80%, 50%, and 200%, respectively). Palmitoleic and tetracosenoic were not affected by any of the treatments under drought conditions. Interestingly, the combined treatments of Com-AMF or Com-AMF-CNPs positively affected the contents of heptadecenoic, linoleic, linolenic, and eicosanoic acids, when compared with drought-stressed plants. 

### 3.7. Polyamine Metabolism

Polyamines could play a key role in oxidative stress by enhancing several antioxidant enzymes. Therefore, we measured the individual polyamines (i.e., putrescine, spermine, and spermidine) and their total in maize plants treated with Com-AMF-CNPs under control or drought conditions. In our study, the treatment with Com did not enhance significant changes in all the measured individual polyamines, as compared with the control plants (Figure 3). In contrast, the AMF treatment significantly enhanced all the individual polyamines, while the treatment with CNPs increased the contents of putrescine and spermidine but did not affect spermine. Moreover, a combination of Com-AMF or Com-nano did not cause significant changes in all the detected polyamines, when compared with the control values. Meanwhile, the triple treatment with Com-AMF-CNPs resulted in remarkable increases in putrescine and spermine (40% and 20%, respectively), while spermidine was not affected (Figure 3). The total polyamines were not affected by any of the treatments, except for the triple treatment with Com-AMF-CNPs which increased the total polyamines by 30% in comparison to the control sample. When drought conditions were applied to maize plants, in addition to the sole treatments with Com or AMF, no significant increases in the individual polyamines were observed. In contrast, all the individual polyamines were remarkably enhanced when plants were treated with CNPs. The combined treatment with Com-AMF-CNPs notably increased the levels of putrescine, spermine, and spermidine (120%, 80%, and 90%, respectively), as compared with the drought-stressed plants (Figure 3).

To explore the mechanism associated with the changes in polyamines under different treatments, we investigated the polyamine-related enzymes involved in the biosynthetic pathway, i.e., arginine decarboxylase, which is incorporated in the biosynthesis of putrescine in plants, and orinthnine decarboxylase, which plays a role in the decarboxylation of ornithine to produce putrescine, which could be converted into spermidine and spermine. We also measured the activity of S-adenosyl-L-methionine decarboxylase, which is involved in the conversion of S-adenosyl methionine into S-adenosylmethioninamine, in addition to its role in the biosynthesis of polyamines by providing the precursors needed for the production of spermidine and spermine from putrescein. In addition, we determined the activity of spermidine (Spd) synthase, which is engaged in the biosynthesis of spermidine from putrescine. In our study, none of the single treatments had a big effect on the activities of arginine decarboxylase or orinthine decarboxylase under control conditions. However, the S-adenosyl-L-methionine decarboxylase activity went up.

Spd synthase was not affected by Com treatments, but it was increased under treatments with AMF or CNPs, as compared with the control samples (Table 6). When Com was combined with AMF, a significant increase was observed in S-adenosyl-L-methionine decarboxylase (by 10%); however, no significant changes in the activities of arginine decarboxylase or Spd synthase were reported. The combined treatment with Com-CNPs induced notable increments in both orinthnine decarboxylase and S-adenosyl-L-methionine decarboxylase (by 40% and 20%, respectively). Moreover, a combination of Com-AMF-CNPs exerted positive effects on orinthnine decarboxylase, S-adenosyl-L-methionine decarboxylase, and Spd synthase (by 100%, 20%, and 30%, respectively). Table 6 shows that under drought conditions, the treatment with Com or AMF did not induce significant effects on all enzyme activities, while the treatment with CNPs only enhanced the activity of Spd synthase (70%). No significant changes were observed when Com was combined with AMF, as compared with the drought-stressed plants. Interestingly, the combined treatment of Com-AMF-CNPs significantly promoted the activities of arginine decarboxylase, S-adenosyl-L-methionine decarboxylase, and Spd synthase (by 90%, 80%, and 70%, respectively). Overall, the combined effect of Com, AMF, and/or CNPs might be more beneficial than their individual treatments to increase the polyamines content and their biosynthetic enzymes.

## 4. Discussion

### 4.1. Drought Conditions Affect Growth and Metabolism

To evaluate the effect of drought, we analyzed the changes in plant growth, physiology, and primary (carbohydrate, lipid, amino acid) and secondary metabolism (polyamine). Under drought stress conditions, maize plants showed reduced growth and photosynthesis. Reduced biomass accumulation and CO_2_ assimilated during photosynthesis reduced C-based metabolites and decreased the production of primary and secondary metabolites under water deficiency conditions [31]. On the other hand, this directed the metabolism to secondary metabolites [32]. For instance, drought-stressed maize plants accumulated more soluble sugars and polyamines to mitigate drought stress in maize plants. Polyamines could play a role in scavenging free radicals [33] and signaling molecules which might be involved in ion transportation [34]. The increased soluble sugars play an effective role in the osmotic adjustment in response to drought stress. Sugars are considered the basic structural components in plants [35], having the ability to scavenge ROS, besides their role in osmoregulation [36] and the production of antioxidant metabolites, all of which end up with improved plant tolerance to unfavorable environmental conditions. Moreover, polyamines are incorporated in the maintenance of membrane stability by the destruction of ROS, which consequently protects the plant from lipid peroxidation induced under stress conditions [37,38,39,40]. Here is an interactive effect of Com, AMF, and CNPs increasing polyamine metabolism (spermine and Spd synthases, ODC, and ADC) in treated maize.

### 4.2. Com, AMF, and CNPs Mitigated Drought Stress Effect on Maize Plant Metabolism

#### 4.2.1. Primary Metabolism

Here, for the first time, we shed light on how Com, AMF, and/or CNPS mitigated drought stress effects on maize plants. Here, Com, AMF, and/or CNPS increased plant growth and photosynthesis. By enhancing the biological and physical conditions of the soil, Com enhances plant growth and yield [41]. One of the most important components of sustainable farming is using biological and organic fertilizers to deliver nutrients to plants while minimizing the demand for chemical fertilizers [42]. Additionally, Com amendment to the soil increased the amount of total chlorophyll in the leaves, perhaps because of the addition of necessary ions to the soil [43,44]. Organic matter releases the macro- and micronutrients required to produce enzymes involved in photochemical reactions over time [44]. This is further improved by inoculation with AMF, which also enhances phytohormone production, increases mineral nutrients in the leaves, and enhances pigment content, photosynthesis, and the antioxidant defense system for eggplant [45]. In addition to their well-known functions in increasing mineral absorption [46], AMF agents and mineral nutrition improved the enzymatic defense, morphophysiological characteristics, quality, and yield of *Glycyrrhiza glabra* under drought stress [43]. On the other hand, AMF may cooperate with other bacterial communities to promote compost waste organic matter decomposition and nutrient absorption [42,43]. Surprisingly, the presence of plant waste compost enhances soil microbial activity, particularly AMF, because it provides a significant amount of organic carbon [47,48]. In agreement with our findings, Schulze et al. [49] and Gao et al. [50] recorded that CNPs increased photosynthesis parameters including RuBisCO activase activity and chlorophyll synthesis, both of which increased biomass accumulation. Additionally, nanoparticles enhance the nutrients produced by composted plant waste and deliver them to plants [51]. Furthermore, AMF colonization was stimulated by nanoparticles [52]. In a similar manner, CNPs can enhance AMF colonization in this situation, increasing soil fertility and plant output as a result. In this regard, Lahiani et al. [53] demonstrated how CNP exposure affected the germination and growth of soybean, corn, and barley. In one of these studies, CDs encouraged the mung bean sprouts to develop larger roots and stems, build more biomass, and boost their photosynthesis and glucose [54,55]. In response to Com, AMF, and/or CNPS treatments during a drought, plants should make more sugars because photosynthesis will be stronger, and respiration will be better. The accumulated sugars and other metabolites could enhance plant growth under stress conditions, in addition to their role as osmoprotectants to alleviate the harmful effects of stress conditions [56]. Our results show enhancements in sucrose metabolism, which might be a mechanism to cope with drought stress. This could also play a role in osmoregulation and providing building blocks for stress-tolerant metabolites [57].

Under stress conditions, the changes in sugar metabolism might also affect the tricarboxylic acid cycle intermediates, e.g., organic acids. In this regard, organic acids could represent the C skeleton needed for the biosynthesis of amino acids, and thus, they could serve as a channel between C and N metabolism [58]. In addition, organic acids could be considered osmoprotectants that protect the plant against stress conditions [59]. Through the control of many physiological, biochemical, and molecular processes, nanoparticles have been shown in numerous studies to play a significant role in protecting plants against abiotic stress. Additionally, during abiotic stress, nanoparticles are typically implicated in boosting the activities of both enzymatic and non-enzymatic antioxidants. The increments in organic acids could positively affect the redox status of the cell, thus providing the precursors needed to produce amino acids. Further, the increased amounts of organic acids might exert a positive impact on the plant’s nutritional values by improving its taste and flavor [60].

Amino acids could be synthesized in plants through sugar breakdown, which provides the C skeletons needed for their biosynthesis. Under stressed conditions, amino acids could also affect N assimilation [61]. When water is scarce, proline accumulates in the roots of several plant species where AM fungi have colonized them [62,63,64]. The increased proline accumulation in these trials was associated with AM-induced drought tolerance, with proline serving as an osmoprotectant. Many previous studies have found that while proline content increased in response to water deficits, AMF-treated plants accumulate less proline than their nonmycorrhizal counterparts, suggesting that the AMF symbiosis increased host plant resistance to drought [65,66,67,68]. In addition to their role in protein synthesis, amino acids could also provide several osmolytes that keep the plant turgor pressure constant [69]. For instance, proline could be involved in protecting cell structure and enzymes, as well as increasing cell turgidity [69]. So, proline could play a key role in enhancing the plant’s defense against stress conditions.

Fatty acids are considered among the tricarboxylic acid cycle intermediates and could be biosynthesized by utilizing sugars as their building blocks. Thus, the changes in sugar metabolism might induce significant effects on fatty acids in response to different treatments with Com, AMF, and/or CNPs under drought conditions. Our results show significant increases in fatty acid contents under treatments with Com or AMF, and in this context, fatty acids could be regarded as important contributors to the production of energy for a variety of physiological processes in plants. Recently, it has been demonstrated that nanoparticles can alter lipid metabolism in plants [70,71]. According to Martinez-Ballesta et al. [19], nanopriming with CNPs improved the transport of water, ions, and aquaporins through cell membranes. According to Al Jaouni et al. [72], vermicompost had a substantial impact on the fatty acid composition and oil content of various cultivars of date palm fruits. AMF colonization improves resource efficiency, fatty acid profile, and yield stability, lowering the production risks associated with crops grown under drought stress conditions. It was decided that AMF colonization should be used to make drought stress less bad [43,73].

#### 4.2.2. Secondary Metabolism

The enhanced levels of amino acids under different treatments could also promote the polyamine content. For example, putrescine could be synthesized from ornithine by ODC or from arginine by ADC [74]. In our study, the total and individual polyamines were remarkably increased in maize plants in response to Com, AMF, and/or CNPs. In this regard, polyamines are known to help the plant overcome metal and metalloid stresses [75]. They protect the plant against lipid peroxidation induced under stress conditions [40]. Further, polyamines play a role in maintaining the nucleic acid’s status and in their translocation inside the plant cell [30]. According to Niemi et al.’s [76] findings, one of the first signs that an ectomycorrhizal association has been established is the buildup of PAs in the host plant. Additionally, there is more proof that PAs are involved in mycorrhizal interactions. AM fungus in the lab has been shown to make and release different PAs depending on the species and strain [77].

## 5. Conclusions

The use of a combination of Com, AMF, and CNPs is a promising strategy that has the potential to protect plants against dry stress. Corn plants cultivated under drought stress benefited from the mixture’s physiological and biochemical features, as well as overall growth and biomass accumulation. These treatments boost secondary metabolism. AMF, in addition to CNPs, play an important role in reducing drought stress. The combination of CNPs, AMF, and Com could be a novel strategy to combat drought stress. The new utilization of CNPs in agriculture could help meet rising food demand while also preserving the environment. However, more research needs to be conducted on how CNPs work, what limits are safe, and how toxic they are to the environment in edible plants.

## Figures and Tables

**Figure 1 plants-11-03324-f001:**
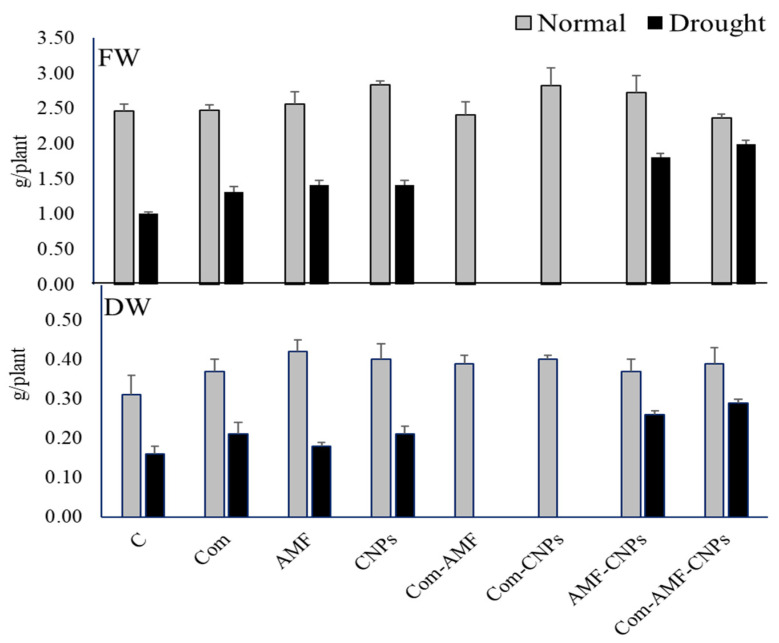
Effect of different treatments on fresh (FW) and dry (DW) weights of maize under control and drought (D) conditions. Different letters indicate statistically significant difference between means for the different treatments at significance level of at least *p* ≤ 0.05. C: control, Com: compost, AMF: mycorrhiza, CNPs: carbon nanoparticles, D: drought.

**Figure 2 plants-11-03324-f002:**
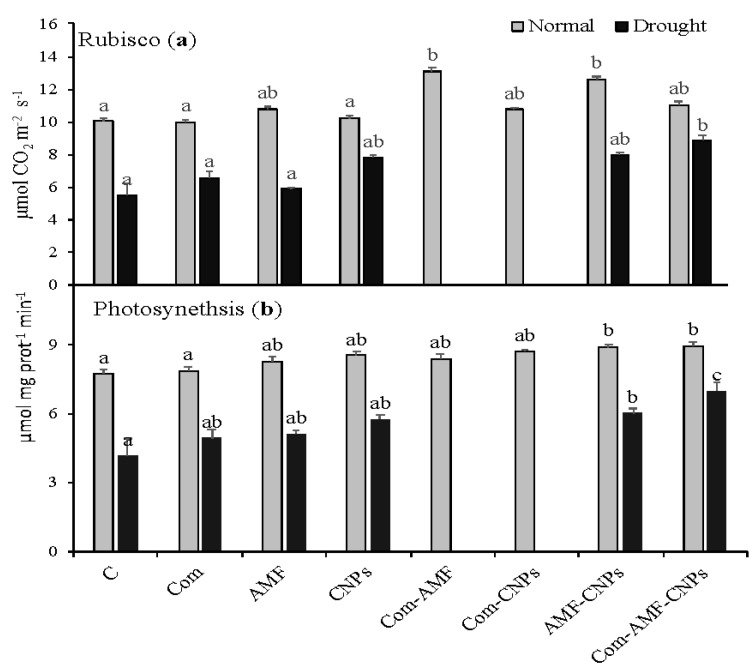
Effect of different treatments on the activity of ribulose-1,5-bisphosphate carboxylase-oxygenase (**a**) and photosynthesis (**b**) of maize under control and drought (D) conditions. Different letters indicate statistically significant difference between means for the different treatments at significance level of at least *p* ≤ 0.05. C: control, Com: compost, AMF: mycorrhiza, CNPs: carbon nanoparticles, D: drought.

**Figure 3 plants-11-03324-f003:**
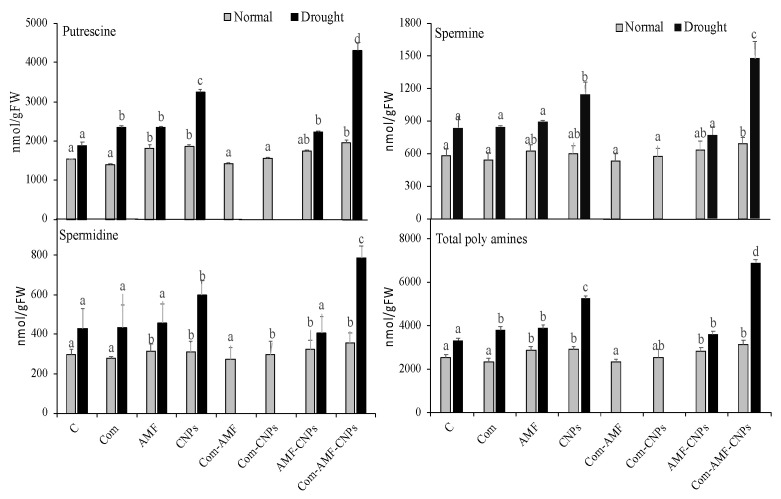
Effect of different treatments on polyamines of maize under control and drought (D) conditions. Different letters indicate statistically significant difference between means for the different treatments at significance level of at least *p* ≤ 0.05.

**Table 1 plants-11-03324-t001:** Effect of different treatments on sugar contents of maize under control and drought (D) conditions. Different letters indicate statistically significant difference between means for the different treatments at significance level of at least *p* ≤ 0.05. C: control, Com: compost, AMF: mycorrhiza, CNPs: carbon nanoparticles, D: drought.

	Glucose mg/g FW	Fructose mg/g FW	Sucrose ug/g FW	Total S Sugars	Sucrose P Synthase	Invertase	Sucrose Synthase
C	2.11 ± 0.03 a	0.86 ± 0.1 a	2.24 ± 0.13 ac	4.18 ± 0.13 c	0.27 ± 0 d	0.6 ± 0.01 d	6.19 ± 0.81 c
Com	2.1 ± 0.01 a	0.96 ± 0.11 a	2.34 ± 0.16 c	4.31 ± 0.09 c	0.26 ± 0 d	0.55 ± 0 d	6.69 ± 0.88 c
AMF	2.64 ± 0.13 a	1.2 ± 0.12 a	2.61 ± 0.22 c	5.38 ± 0.17 b	0.32 ± 0 c	0.61 ± 0.03 d	6.96 ± 0.71 c
CNPs	2.59 ± 0.01 a	1.09 ± 0.13 a	2.29 ± 0.13 c	5.12 ± 0.12 b	0.25 ± 0 d	0.65 ± 0 d	4.78 ± 0.63 e
Com-AMF	2.28 ± 0.02 a	1.19 ± 0.14 a	2.81 ± 0.23 b	4.92 ± 0.09 c	0.26 ± 0 d	0.56 ± 0 d	6.78 ± 0.89 d
Com-CNPs	2.5 ± 0.03 a	1.17 ± 0.14 a	3.2 ± 0.06 b	5.55 ± 0.23 b	0.42 ± 0 c	0.9 ± 0.01 c	7.32 ± 0.98 d
AMF-CNPs	2.21 ± 0.02 a	1.03 ± 0.12 a	2.6 ± 0.13 c	4.67 ± 0.12 c	0.31 ± 0 c	0.68 ± 0 d	7.73 ± 1.02 d
Com-AMF-CNPs	2.1 ± 0.05 a	0.98 ± 0.11 a	2.93 ± 0.07 c	4.95 ± 0.24 c	0.34 ± 0 c	1.12 ± 0.02 c	6.26 ± 0.8 c
D	1.9 ± 0.04 a	0.86 ± 0.1 a	3.06 ± 0.12 b	4.75 ± 0.35 c	0.5 ± 0 b	1.41 ± 0.03 b	6.21 ± 0.79 c
D-Com	2.33 ± 0.08 a	1.43 ± 0.21 b	4.27 ± 0.28 a	6.52 ± 0.63 a	0.55 ± 0.02 b	1.6 ± 0.06 a	6.56 ± 0.84 c
D-AMF	1.9 ± 0.02 a	1.22 ± 0.15 a	4.06 ± 0.15 b	5.77 ± 0.43 b	0.59 ± 0 b	1.82 ± 0.02 a	6.39 ± 0.84 c
D-CNPs	2.56 ± 0.03 a	1.14 ± 0.11 a	3.89 ± 0.14 b	6.13 ± 0.32 a	0.6 ± 0.02 b	1.4 ± 0.1 b	8.59 ± 1.42 b
D-Com-AMF	2.67 ± 0.05 a	1.2 ± 0.13 a	4.4 ± 0.2 a	6.78 ± 0.53 a	0.86 ± 0.01 a	2.14 ± 0.05 a	12.52 ± 1.61 a
D-Com-AMF-CNPs	2.82 ± 0.01 a	1.47 ± 0.18 b	4.67 ± 0.07 a	7.18 ± 0.42 a	0.7 ± 0 b	1.8 ± 0.01 a	13.05 ± 1.72 a

**Table 2 plants-11-03324-t002:** Effect of different treatments on organic acids (mg g^−1^ DW) of maize under control and drought (D) conditions. Different letters indicate statistically significant difference between means for the different treatments at significance level of at least *p* ≤ 0.05.

	Oxalic Acid	Malic Acid	Succinic Acid	Citric Acid	Isobutyric Acid	Fumaric Acid
C	2.36 ± 0.03 a	9.41 ± 0.1 b	2.33 ± 0.04 b	1.25 ± 0.02 d	3.92 ± 0.07 a	0.31 ± 0 a
Com	1.97 ± 0.02 a	7.25 ± 0.08 c	1.96 ± 0.02 b	3.27 ± 0.03 b	2.69 ± 0.03 b	0.27 ± 0 a
AMF	2.4 ± 0.02 a	10.7 ± 0.13 b	1.92 ± 0.02 b	2.37 ± 0.04 c	3.49 ± 0.04 a	0.27 ± 0 a
CNPs	1.11 ± 0.02 b	13.24 ± 0.14 a	2.04 ± 0.08 b	4.41 ± 0.05 a	2.64 ± 0.03 b	0.26 ± 0 a
Com-AMF	2.94 ± 0.03 a	14.76 ± 0.16 a	1.33 ± 0.28 c	1.28 ± 0.02 d	1.93 ± 0.02 b	0.21 ± 0 a
Com-CNPs	2.23 ± 0.02 a	11.06 ± 0.12 b	1.29 ± 0.02 c	2.71 ± 0.03 c	3.2 ± 0.03 a	0.22 ± 0 a
AMF-CNPs	2.25 ± 0.02 a	10.59 ± 0.11 b	2.04 ± 0.08 b	1.3 ± 0.02 d	3.45 ± 0.14 a	0.27 ± 0 a
Com-AMF-CNPs	2.35 ± 0.03 a	10.38 ± 0.11 b	2.12 ± 0.08 b	1.35 ± 0.02 d	3.6 ± 0.15 a	0.29 ± 0 a
D	1.19 ± 0.02 b	14.1 ± 0.15 a	1.74 ± 0.05 b	2.47 ± 0.02 c	2.81 ± 0.03 b	0.28 ± 0 a
D-Com	1.98 ± 0.02 a	16.9 ± 0.18 a	1.63 ± 0.06 b	5.36 ± 0.06 a	2.81 ± 0.03	0.28 ± 0 a
D-AMF	2.46 ± 0.03 a	12.37 ± 0.13 b	1.7 ± 0.04 b	2.6 ± 0.05 c	2.73 ± 0.11 b	0.26 ± 0 a
D-CNPs	1.26 ± 0.05 b	9.53 ± 0.18 b	0.91 ± 0.04 c	1.04 ± 0.01 d	2.26 ± 0.11 b	0.19 ± 0 a
D-Com-AMF	2.68 ± 0.03 a	11.59 ± 0.12 b	1.44 ± 0.02 b	2.79 ± 0.03 c	3.27 ± 0.04 a	0.22 ± 0 a
D-Com-AMF-CNPs	2.22 ± 0.13 a	12.3 ± 0.76 b	3.34 ± 0.13 a	2.14 ± 0.03 c	2.38 ± 0.12 b	0.28 ± 0.02 a

**Table 3 plants-11-03324-t003:** Effect of different treatments on amino acids (mg g^−1^ DW) of maize under control and drought (D) conditions. Different letters indicate statistically significant difference between means for the different treatments at significance level of at least *p* ≤ 0.05.

	Glycine	Lysine	Histidine	Alanine	Arginie	Ornithine	Glutamine	Asparagine	Isoleucine	Leucine	Mean	Threonine	Valine	Serine	Phenylalanine	GlutamIic acid	Aspartate	Cystine	Mean
C	59.02 ± 0.38 b	3.61 ± 0.05 c	1.39 ± 0.02 b	4.43 ± 0.03 d	0.55 ± 0.01 b	0.28 ± 0.02 c	0.83 ± 0.03 c	0.66 ± 0 c	0.1 ± 0 d	0.03 ± 0 d	0.05 ± 0 d	0.23 ± 0.02 e	0.38 ± 0 c	0.41 ± 0.03 b	0.59 ± 0 d	0.11 ± 0 e	0.06 ± 0.03 c	0.16 ± 0.02	0.36 ± 0 c
Com	39.08 ± 0.25 c	2.81 ± 0.03 d	1.59 ± 0.08 b	11.86 ± 0.08 b	1.44 ± 0.1 a	0.26 ± 0.01 d	0.36 ± 0.01 d	1.23 ± 0.01 a	0.17 ± 0 d	0.03 ± 0 d	0.08 ± 0.01 c	0.48 ± 0.04 c	0.81 ± 0.01 a	0.29 ± 0.01 c	0.47 ± 0.03 d	0.61 ± 0 c	0.09 ± 0 c	0.22 ± 0.02 c	0.38 ± 0 c
AMF	43.11 ± 0.4 c	4.6 ± 0.04 b	1.72 ± 0.05 a	18.69 ± 0.15 a	1.47 ± 0.05 a	0.45 ± 0.05 b	1.55 ± 0.07 b	0.79 ± 0.06 c	0.48 ± 0 b	0.07 ± 0 c	0.07 ± 0.01 c	0.46 ± 0.07 c	0.75 ± 0.03 a	0.51 ± 0.03 a	0.71 ± 0.05 c	0.8 ± 0.01 c	0.14 ± 0.01 b	0.37 ± 0.05 b	0.61 ± 0.01 a
CNPs	35.18 ± 0.23 c	2.47 ± 0.05 d	1.78 ± 0.05 a	6.21 ± 0.05 c	1.24 ± 0.01 a	0.52 ± 0.04 b	1.51 ± 0.09 b	2.28 ± 0.03 a	0.8 ± 0.01 a	0.29 ± 0.01 a	0.41 ± 0.02 a	0.66 ± 0.02 b	0.43 ± 0 b	0.31 ± 0.02 c	0.88 ± 0.05 c	0.52 ± 0.04 d	0.23 ± 0.01 b	0.56 ± 0.04 a	0.72 ± 0.01 a
Com-AMF	43.9 ± 0.3 c	6.5 ± 0.04 a	1.59 ± 0.07 b	9.66 ± 0.07 b	1.48 ± 0.01 a	0.29 ± 0.01 c	1.22 ± 0.01 b	0.73 ± 0.01 c	0.18 ± 0 d	0.04 ± 0 d	0.04 ± 0 d	0.17 ± 0.01 e	0.52 ± 0.04	0.24 ± 0 c	0.44 ± 0.04 d	0.72 ± 0 c	0.08 ± 0 c	0.38 ± 0.06 b	0.78 ± 0.01 a
Com-CNPs	73.3 ± 0.47 a	4.27 ± 0.05 c	1.76 ± 0.05 a	12.87 ± 0.08 b	1.11 ± 0.01 a	0.39 ± 0.01 c	0.88 ± 0.02 c	0.63 ± 0.01 c	0.2 ± 0 d	0.09 ± 0 c	0.08 ± 0 d	0.37 ± 0.03 d	0.71 ± 0.01 a	0.52 ± 0.03 a	0.52 ± 0.04 d	0.52 ± 0.04 d	0.16 ± 0 b	0.45 ± 0.06 b	0.91 ± 0.01 a
AMF-CNPs	48.44 ± 0.31 b	3.4 ± 0.04 c	1.24 ± 0.02 b	7.57 ± 0.05 c	1.55 ± 0.01 a	0.33 ± 0.01 c	1.36 ± 0 b	0.5 ± 0.01 c	0.28 ± 0.01 c	0.37 ± 0.01 a	0.22 ± 0 b	0.34 ± 0.04 d	0.75 ± 0.01 b	0.32 ± 0.01 b	0.71 ± 0.04 c	0.88 ± 0 c	0.11 ± 0 c	0.32 ± 0.04 b	0.62 ± 0 a
Com-AMF-CNPs	54.62 ± 0.35 b	3.72 ± 0.06 c	1.94 ± 0.05 a	5.97 ± 0.04 d	1.11 ± 0.01 a	0.43 ± 0.01 b	2.78 ± 0.19 a	0.71 ± 0 c	0.29 ± 0.01 c	0.37 ± 0.01 a	0.51 ± 0.03 a	0.24 ± 0.03 e	0.52 ± 0.04 b	0.29 ± 0.01 c	0.66 ± 0.05 c	0.95 ± 0.01 c	0.11 ± 0 c	0.29 ± 0.03 c	0.54 ± 0 b
D	68.44 ± 0.43 a	5.45 ± 0.08 b	2.25 ± 0.03 a	16.78 ± 0.11 a	0.76 ± 0.05 b	0.61 ± 0.01 a	0.76 ± 0.02 c	0.57 ± 0.01 c	0.24 ± 0 c	0.12 ± 0 b	0.11 ± 0.02 c	0.66 ± 0.07 b	0.52 ± 0.04 b	0.62 ± 0.03 a	1.39 ± 0.06 b	1.4 ± 0.01 c	0.2 ± 0 b	0.43 ± 0.06 b	0.88 ± 0.01 a
D-Com	72.01 ± 0.48 a	5.78 ± 0.04 b	1.33 ± 0.02 b	17.52 ± 0.12 a	1.28 ± 0	0.58 ± 0.01 a	0.76 ± 0.02 c	0.57 ± 0.01 c	0.24 ± 0 c	0.12 ± 0 d	0.11 ± 0 c	0.66 ± 0.07 b	1.28 ± 0.01 a	0.62 ± 0.03 a	1.7 ± 0.09 a	2 ± 0.01 a	0.33 ± 0.01 a	0.43 ± 0.06 b	0.88 ± 0.01 a
D-AMF	62.67 ± 0.42 a	3.26 ± 0.08 c	2.79 ± 0.07 a	10.93 ± 0.08 b	0.81 ± 0.01 b	0.82 ± 0.05 a	0.76 ± 0.01 c	0.92 ± 0.02 b	0.42 ± 0 b	0.37 ± 0.01 a	0.53 ± 0.03 a	0.95 ± 0.03 a	0.85 ± 0.01 a	0.51 ± 0.02 a	1.09 ± 0.06 b	1.32 ± 0.01 c	0.25 ± 0.01 b	0.52 ± 0.04 a	0.69 ± 0 a
D-CNPs	50.74 ± 0.1 b	6.25 ± 0.12 a	1.44 ± 0.03 b	11.59 ± 0.1 b	0.45 ± 0.02 b	0.42 ± 0.01 b	1.07 ± 0.04 b	1.02 ± 0.09 b	0.15 ± 0 d	0.29 ± 0 a	0.13 ± 0 c	0.49 ± 0.04 c	0.87 ± 0.01 a	0.31 ± 0.03 c	1.26 ± 0.12 b	1.71 ± 0.05 a	0.2 ± 0 b	0.5 ± 0.05 a	0.57 ± 0 b
D-Com-AMF	54.94 ± 0.35 b	5.74 ± 0.05 b	1.31 ± 0.02 b	12.37 ± 0.08 b	0.58 ± 0.03 b	0.52 ± 0.08 a	1.37 ± 0.09 b	1.81 ± 0.01 a	0.14 ± 0 d	0.12 ± 0 d	0.14 ± 0.01 c	0.57 ± 0.04 c	0.89 ± 0.01 a	0.63 ± 0.05 a	1.73 ± 0.06 a	1.5 ± 0.01 a	0.24 ± 0 b	0.39 ± 0.03 b	0.62 ± 0 a
D-Com-AMF-CNPs	62.24 ± 3.34 a	6.93 ± 0.42 a	2.18 ± 0.01 a	18.91 ± 0.13 a	0.66 ± 0.06 b	0.67 ± 0.08 a	1.19 ± 0.05 b	1.33 ± 0.08 a	0.11 ± 0 d	0.24 ± 0 b	0.11 ± 0	0.5 ± 0.05 c	0.95 ± 0.05 a	0.55 ± 0.06 a	2.02 ± 0.07 a	2.08 ± 0.11 a	0.3 ± 0.02 a	0.64 ± 0.08 a	0.61 ± 0.04 a

**Table 4 plants-11-03324-t004:** Effect of different treatment proline metabolic enzymes of maize under control and drought (D) conditions. Different letters indicate statistically significant difference between means for the different treatments at significance level of at least *p* ≤ 0.05.

	Proline	P5CS	P5CR	OAT	PRODH
C	2.23 ± 0.06 c	2.73 ± 0.04 b	0.41 ± 0.01 b	3.56 ± 0.06 d	5.42 ± 0.09 c
Com	2.43 ± 0.11 c	2.73 ± 0.02 b	0.41 ± 0.02 b	3.54 ± 0.03 d	5.47 ± 0.04 c
AMF	2.92 ± 0.14 c	2.88 ± 0.12 b	0.43 ± 0.02 b	3.77 ± 0.16 d	5.74 ± 0.23 c
CNPs	2.98 ± 0.12 c	2.39 ± 0.01 b	0.26 ± 0.01 c	2.72 ± 0.01 d	4.05 ± 0.02 d
Com-AMF	2.25 ± 0.1 c	2.25 ± 0.02 c	0.39 ± 0.02 b	3.58 ± 0.02 d	5.52 ± 0.04 c
Com-CNPs	2.26 ± 0.1 c	2.57 ± 0.03 b	0.43 ± 0.02 b	4.13 ± 0.04 c	5.99 ± 0.06 c
AMF-CNPs	2.69 ± 0.12 c	2.23 ± 0.02 c	0.44 ± 0.02 b	4.12 ± 0.03 c	6.33 ± 0.05 c
Com-AMF-CNPs	2.72 ± 0.16 c	2.48 ± 0.05 b	0.37 ± 0.02 b	3.52 ± 0.08 d	4.84 ± 0.1 c
D	5.15 ± 0.12 a	3.64 ± 0.07 b	0.45 ± 0.01 b	4.05 ± 0.08 c	5.52 ± 0.1 c
D-Com	5.15 ± 0.11 a	3.37 ± 0.11 b	0.48 ± 0.01 b	4.47 ± 0.15 c	6.25 ± 0.21 c
D-AMF	6.01 ± 0.27 a	3.86 ± 0.03 b	0.44 ± 0.02 b	4.13 ± 0.04 c	5.28 ± 0.05 c
D-CNPs	5.01 ± 0.04 b	3.66 ± 0.15 b	0.57 ± 0.01 b	5.92 ± 0.44 c	8.47 ± 0.64 b
D-Com-AMF	6.74 ± 0.14 a	4.72 ± 0.1 a	0.81 ± 0.02 a	9.27 ± 0.19 a	13.3 ± 0.27 a
D-Com-AMF-CNPs	7.04 ± 0.23 a	5.77 ± 0.02 a	0.85 ± 0.03 a	7.5 ± 0.03 b	10.86 ± 0.05 b

**Table 5 plants-11-03324-t005:** Effect of different treatments on fatty acids (µg/g FW) of maize under control and drought (D) conditions. Different letters indicate statistically significant difference between means for the different treatments at significance level of at least *p* ≤ 0.05.

	Myristic (C14:0)	Palmitic (C16:0)	Heptadecanoic (C17:0)	Stearic (C18:0)	Arachidic (C20:0)	Docosanoic (C22:0)	Tricosanoic (C23:0)	Pentacosanoic (C25:0)	Palmitoleic (C16:1)	Heptadecenoic (C17:1)	Oleic (C18:1)	Linoleic (C18:2)	Linolenic (C18:3 ω−3)	Eicosenoic (C20:1)	Eicosenoic (C20:1)	Tetracosenoic (C24:1)
C	0.34 ± 0 c	19.95 ± 0.44 c	0.03 ± 0 b	1.19 ± 0.02 c	1.16 ± 0.02 b	0.52 ± 0.01 c	0.022 ± 0 c	0.01 ± 0 c	0.08 ± 0 c	0.15 ± 0.01 c	37.03 ± 0.31 c	0.014 ± 0 c	0.11 ± 0.02 c	0.91 ± 0.03 b	0.06 ± 0 b	0.01 ± 0 a
Com	0.4 ± 0 b	19.87 ± 0.1 c	0.03 ± 0 b	1.54 ± 0.03 b	1.16 ± 0.01 b	0.55 ± 0.01 c	0.027 ± 0 c	0.03 ± 0 c	0.07 ± 0 c	0.15 ± 0.01 c	33.7 ± 0.54 c	0.021 ± 0 b	0.021 ± 0 b	0.9 ± 0.01 b	0.06 ± 0 b	0.02 ± 0
AMF	0.46 ± 0.02 b	19.12 ± 0.62 c	0.04 ± 0 b	1.47 ± 0.04 b	1.46 ± 0.07 b	0.68 ± 0.04 b	0.034 ± 0 c	0.02 ± 0 c	0.07 ± 0 c	0.16 ± 0.01 c	43.88 ± 2.23 c	0.02 ± 0 b	0.02 ± 0 b	0.97 ± 0.01 b	0.06 ± 0 b	0.016 ± 0 a
CNPs	0.5 ± 0 b	17.88 ± 0.28 c	0.03 ± 0 b	1.81 ± 0.04 b	1.43 ± 0.01 b	0.64 ± 0.01 b	0.036 ± 0 c	0.02 ± 0 c	0.08 ± 0 c	0.11 ± 0.01 c	40.1 ± 0.4 c	0.018 ± 0 b	0.013 ± 0 c	0.51 ± 0.03 c	0.04 ± 0 c	0.015 ± 0 a
Com-AMF	0.36 ± 0 c	17.06 ± 0.14 c	0.03 ± 0 b	1.5 ± 0.04 b	1.26 ± 0.01 b	0.69 ± 0.01 b	0.027 ± 0 c	0.01 ± 0 c	0.07 ± 0 c	0.15 ± 0.01 c	34.4 ± 0.55 c	0.022 ± 0 b	0.022 ± 0 b	0.98 ± 0.02 b	0.06 ± 0 b	0.016 ± 0 a
Com-CNPs	0.35 ± 0 c	18.52 ± 0.18 c	0.07 ± 0 a	1.37 ± 0.02 b	1.38 ± 0.01 b	0.62 ± 0.03 b	0.05 ± 0 b	0.01 ± 0 c	0.1 ± 0 b	0.17 ± 0.01 c	37.36 ± 0.65 c	0.016 ± 0 b	0.02 ± 0 b	1.13 ± 0.01 b	0.07 ± 0 b	0.017 ± 0 a
AMF-CNPs	0.43 ± 0 b	16.81 ± 0.21 c	0.04 ± 0 b	1.7 ± 0.04 b	1.22 ± 0.01 b	0.59 ± 0.01 c	0.034 ± 0 b	0.03 ± 0 c	0.08 ± 0 c	0.18 ± 0.01 b	42.09 ± 0.67 c	0.02 ± 0 b	0.021 ± 0 a	1.11 ± 0.05 b	0.07 ± 0 b	0.018 ± 0 a
Com-AMF-CNPs	0.33 ± 0 c	19.22 ± 0.16 c	0.07 ± 0 a	1.21 ± 0.03 c	1.15 ± 0.02 c	0.52 ± 0.01 c	0.05 ± 0 b	0.02 ± 0 c	0.13 ± 0.01 b	0.14 ± 0.01 c	50.45 ± 3.64 b	0.021 ± 0 b	0.023 ± 0 b	1.3 ± 0.06 b	0.05 ± 0 b	0.018 ± 0 a
D	0.58 ± 0.01 b	30.15 ± 0.88 b	0.08 ± 0 a	2.11 ± 0.03 a	1.03 ± 0.02 c	0.52 ± 0.01 c	0.06 ± 0 b	0.012 ± 0 b	0.18 ± 0 a	0.15 ± 0 c	52.4 ± 2.41 b	0.02 ± 0 b	0.015 ± 0 b	0.99 ± 0.03 b	0.06 ± 0 b	0.017 ± 0 a
D-Com	1.02 ± 0.01 a	30.64 ± 1.05 b	0.11 ± 0 a	2.5 ± 0.04 a	1.04 ± 0.01 c	0.7 ± 0.01 b	0.07 ± 0.01 a	0.017 ± 0 b	0.22 ± 0.01 a	0.14 ± 0.01 c	60.3 ± 0.61 a	0.012 ± 0 c	0.017 ± 0 b	1.02 ± 0.04 b	0.06 ± 0 b	0.017 ± 0 a
D-AMF	1.15 ± 0.04 a	33.77 ± 1.83 b	0.11 ± 0.01 a	2.98 ± 0.07 a	1.21 ± 0.03 b	0.88 ± 0.03 b	0.08 ± 0.01 a	0.014 ± 0 b	0.17 ± 0 a	0.19 ± 0.01 b	64.18 ± 3.78 a	0.019 ± 0 b	0.02 ± 0 b	1.23 ± 0.03 b	0.08 ± 0 b	0.02 ± 0 a
D-CNPs	0.91 ± 0.04 a	33.58 ± 1.62 b	0.12 ± 0.01 a	2.35 ± 0.1 a	2 ± 0.1 a	0.99 ± 0.05 a	0.07 ± 0 a	0.021 ± 0 a	0.2 ± 0.01 a	0.25 ± 0 a	85.29 ± 5.56 a	0.035 ± 0 a	0.03 ± 0 a	1.57 ± 0.04 b	0.12 ± 0 a	0.02 ± 0 a
D-Com-AMF	0.95 ± 0.01 a	36.2 ± 0.4 b	0.13 ± 0 a	3.01 ± 0.04 a	1.43 ± 0.03 b	0.99 ± 0.08 a	0.09 ± 0 a	0.021 ± 0 a	0.21 ± 0.01 a	0.3 ± 0.01 a	51.05 ± 8.84 b	0.046 ± 0 a	0.032 ± 0 a	2.31 ± 0.08 a	0.14 ± 0 a	0.03 ± 0 a
D-Com-AMF-CNPs	1.11 ± 0.04 a	42.73 ± 0.78 a	0.1 ± 0 a	3.1 ± 0.1 a	2.13 ± 0.01 a	1.36 ± 0.08 a	0.059 ± 0 b	0.033 ± 0 a	0.17 ± 0 a	0.28 ± 0.01 a	55.24 ± 1.95 b	0.037 ± 0 a	0.037 ± 0 a	1.81 ± 0.08 a	0.11 ± 0 a	0.03 ± 0 a

**Table 6 plants-11-03324-t006:** Effect of different treatments on polyamine-related enzymes of maize under control and drought (D) conditions. Different letters indicate statistically significant difference between means for the different treatments at significance level of at least *p* ≤ 0.05.

	Orinthnine Decarboxylase (µ mol/mg Protein min)	S-adenosyl-L-Methionine Decarboxylase (µ mol/mg Protein min)	Spermidine (Spd) Synthase (µ mol/mg Protein min)	Spd Synthase (µ mol/mg Protein min)
C	0.12 ± 0 c	13.28 ± 0.28 d	38.64 ± 0.29 c	17.99 ± 0.06 c
Com	0.1 ± 0 c	18.73 ± 0.14 c	35.23 ± 0.55 c	16.54 ± 0.13 c
AMF	0.12 ± 0 c	18.12 ± 0.62 c	41.92 ± 0.61 b	19.06 ± 0.1 c
CNPs	0.13 ± 0 c	17.13 ± 0.25 c	41.5 ± 0.41 b	18.54 ± 0.08 c
Com-AMF	0.11 ± 0 b	16.21 ± 0.12 c	36.1 ± 0.56 c	16.34 ± 0.11 c
Com-CNPs	0.17 ± 0 a	17.56 ± 0.18 c	39.23 ± 0.64 c	17.75 ± 0.15 c
AMF-CNPs	0.13 ± 0 b	17.78 ± 0.46 c	43.9 ± 0.7 b	19.38 ± 0.18 c
Com-AMF-CNPs	0.21 ± 0.01 a	18.36 ± 0.13 c	48.83 ± 1.7 b	21.74 ± 0.36 c
D	0.28 ± 0 a	27.68 ± 0.47 b	55.82 ± 0.49 b	26 ± 0.08 c
D-Com	0.35 ± 0.01 a	24.03 ± 0.44 c	58.93 ± 1.25 b	26.03 ± 0.29 c
D-AMF	0.22 ± 0.01	27.98 ± 1.41 b	58.9 ± 0.54 b	27.66 ± 0.05 c
D-CNPs	0.33 ± 0.01 a	32.33 ± 1.55 b	81.57 ± 1.9 a	36.27 ± 1.0 b
D-Com-AMF	0.34 ± 0.01 a	35.44 ± 0.33 b	38.95 ± 1.04 c	24.63 ± 0.3 c
D-Com-AMF-CNPs	0.37 ± 0.02 a	49.42 ± 2.6 a	81.06 ± 2.97 a	47.6 ± 2.2 a

## Data Availability

Not applicable.

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
