# Peer review of "How Carbon Nanoparticles, Arbuscular Mycorrhiza, and Compost Mitigate Drought Stress in Maize Plant: A Growth and Biochemical Study"

_plants, 2022, doi:10.3390/plants11233324_

Round 1

Reviewer 1 Report

Dear Author:

Please rewrite the introduction because it lacks flow. the importance of the study was not projected.

Where the MWCNT was purchased or how it is synthesized.

Please explain the ultrastructure of MWCNT, any functionalization was done to increase water uptake.

Why primmed and unprimed seeds were sown in the same ridge, it will cause the variation.

What is the significance of bulk density of composite, 

Which organism was added for making composite, as per your procedure compost was made without organism, that means it is dried not compost.

The major thing to be explained is, why interaction was studied, and what is the logic for that. Please explain in detail.

The crop husbandry is missing, and please add that.

The data is from a single experiment; how confident you are that this result will not vary in the next experiment?

Spelling check - throughout the manuscript is needed. Example: Rubisco, this is the correct spelling.

To me, the design of the experiment needs a revisit, to why there is no drought treatment for combinations.

Figure - Please redraw - Stretched

The methodology of Rubisco is missing

HPLC method may be explained better.

Author Response

Please rewrite the introduction because it lacks flow. the importance of the study was not projected.

Response: The introduction has been carefully rewritten.

Where the MWCNT was purchased or how it is synthesized.

Response: The introduction has been rewritten.

Please explain the ultrastructure of MWCNT, any functionalization was done to increase water uptake.

Response: Water-dispersible Carbon nanoparticles were used. Detailed characterization of CNPs are added to the methods section.

Why primmed and unprimed seeds were sown in the same ridge, it will cause the variation.

Response: In the experiment, it was taken into consideration that the control plants are at a large distance from the treated plants. Moreover, a barrier was placed inside the soil to separate the control plants from the treated plants. This step is added in methods.

What is the significance of bulk density of composite

Response: The bulk density is a parameter that characterizes the ratio between the solid and gaseous content in the composite.  It also an indicator of composite homogeneity. The bulk density is a function of the local pressure and depends on the position in the vegetable waste pile.

Which organism was added for making composite, as per your procedure compost was made without organism, that means it is dried not compost.

Response: Phanerochaete chrysosporium  and Trichoderma spp. were added to composite. More information about compost formation is added to methodology section.

The major thing to be explained is, why interaction was studied, and what is the logic for that. Please explain in detail.

Response: The growth promoting effect of individual treatments of CNPs, AMF, and compost is well known. In this context, AMF plays a key role in the soil environment by channeling nutrients of plant waste Comp from the soil to their host plant, resulting in improve plant growth particularity under environmental stress. However, how CNPs can improve the interactive effect of AMF and Comp on corn crops under drought conditions is the main target of this study. Such information might be useful in creating appropriate treatments to increase the tolerance of plants to drought stress.

The crop husbandry is missing, and please add that.

Response: Details about the crop husbandry are added to methodology section.

The data is from a single experiment; how confident you are that this result will not vary in the next experiment?

Our response: First, a preliminary experiment at small scale was done to check the growth of used maize variety under different treatment and stress conditions.  The results of the preliminary experiment were confirmed by the main experiment. Moreover, we are confident about the experiment results due to the enough number of replicates (5) and perform the necessary statistical analyses for them.

Spelling check - throughout the manuscript is needed. Example: Rubisco, this is the correct spelling.

Response: Spelling check - throughout the manuscript has been done

To me, the design of the experiment needs a revisit, to why there is no drought treatment for combinations.

Our response: The main aim of the study was to investigate the interactions among the three treatments (CNP, AMF, and Comp) under normal and drought conditions.

Figure - Please redraw – Stretched

Response: All figures have been redrawn.

The methodology of Rubisco is missing

Response: Added

HPLC method may be explained better.

Response: explained

Reviewer 2 Report

Dear Authors

The purpose of this research is to investigate how the interactive potential of compost (Com), AMF and carbon nanoparticles (CNPs) is applied as a new approach to mitigate drought stress in maize plants. To this end, the plant's, photosynthesis, primary metabolism (sugar content, fatty acids, amino acids) and polyamine were evaluateed. Com, AMF and CNPs, separately or in combination, affect maize development under drought stress. Water deficit was reduced by Com, AMF and/or CNPs.

- line 46 “Due to the limited soil water content, drought stress limits plant growth, development, and productivity including leaf wilting, a decrease in plant height, and biomass accumulation during its early vegetative stage [3-4].”

[4] Cataldo, E., Fucile, M., & Mattii, G. B. (2022). Leaf Eco-Physiological Profile and Berries Technological Traits on Potted Vitis vinifera L. cv Pinot Noir Subordinated to Zeolite Treatments under Drought Stress. Plants, 11(13), 1735.

- line 60 is not clear. “For a long time, organic waste recycling for plant nutrients has been thought to maintain the plant nutrients, keep the soil healthy, keep pollution down, and create jobs”

- line 70-71 “The addition of fertilizers and organic manures comprises mostly nutrients and minerals to enrich the soil quality, and improve the fertility of the soil, and the plant's efficiency [13-15]”.

[15] Cataldo, E. C., Salvi, L. S., Paoli, F. P., Fucile, M. F., Masciandaro, G. M., Manzi, D. M., ... & Mattii, G. B. M. (2021). Effects of natural clinoptilolite on physiology, water stress, sugar, and anthocyanin content in Sanforte (Vitis vinifera L.) young vineyard. The Journal of Agricultural Science, 159(7-8), 488-499.

- line 74 rephrase it “the negative effects of nutrients”

- line 79

[17] Jeffries, P., Gianinazzi, S., Perotto, S., Turnau, K., & Barea, J. M. (2003). The contribution of arbuscular mycorrhizal fungi in sustainable maintenance of plant health and soil fertility. Biology and fertility of soils, 37(1), 1-16.

[18] Jeffries, P., & Barea, J. M. (2012). 4 Arbuscular Mycorrhiza: A Key Component of Sustainable Plant–Soil Ecosystems. Fungal associations, 51-75.

 - line 92 “To this end, the plant's, physiology (photosynthesis), primary metabolism (sugar content, fatty acids, amino acids) and secondary metabolism (polyamine)”.

- line 99 more details 2.1. Compost Formation

- Add information in the experimental design (repetitions, times, stages, etc.)

- tables 3-5 the reading is not clear

Author Response

- line 46 “Due to the limited soil water content, drought stress limits plant growth, development, and productivity including leaf wilting, a decrease in plant height, and biomass accumulation during its early vegetative stage [3-4].”

[4] Cataldo, E., Fucile, M., & Mattii, G. B. (2022). Leaf Eco-Physiological Profile and Berries Technological Traits on Potted Vitis vinifera L. cv Pinot Noir Subordinated to Zeolite Treatments under Drought Stress. Plants, 11(13), 1735.

Response: We added the reference (no. 2 in the reference list).

- line 60 is not clear. “For a long time, organic waste recycling for plant nutrients has been thought to maintain the plant nutrients, keep the soil healthy, keep pollution down, and create jobs”

Response: The sentences changed to “It is reported that recycling organic waste conserves plant nutrients, keeps soil healthy, reduces pollution, and creates jobs”.

- line 70-71 “The addition of fertilizers and organic manures comprises mostly nutrients and minerals to enrich the soil quality, and improve the fertility of the soil, and the plant's efficiency [13-15]”.

[15] Cataldo, E. C., Salvi, L. S., Paoli, F. P., Fucile, M. F., Masciandaro, G. M., Manzi, D. M., ... & Mattii, G. B. M. (2021). Effects of natural clinoptilolite on physiology, water stress, sugar, and anthocyanin content in Sanforte (Vitis vinifera L.) young vineyard. The Journal of Agricultural Science, 159(7-8), 488-499.

Response: We added the reference (no. 8 in the reference list).

- line 74 rephrase it “the negative effects of nutrients”

Our response: we rephrased it.

- line 79

[17] Jeffries, P., Gianinazzi, S., Perotto, S., Turnau, K., & Barea, J. M. (2003). The contribution of arbuscular mycorrhizal fungi in sustainable maintenance of plant health and soil fertility. Biology and fertility of soils, 37(1), 1-16.

Our response: we added the reference (no. 14 in the reference list).

[18] Jeffries, P., & Barea, J. M. (2012). 4 Arbuscular Mycorrhiza: A Key Component of Sustainable Plant–Soil Ecosystems. Fungal associations, 51-75.

Our response: we added the reference (no. 15 in the reference list).

 - line 92 “To this end, the plant's, physiology (photosynthesis), primary metabolism (sugar content, fatty acids, amino acids) and secondary metabolism (polyamine)”.

Our response: we rephrased it.

- line 99 more details 2.1. Compost Formation

Response: We added compost information

- Add information in the experimental design (repetitions, times, stages, etc.)

 Response: Added

- tables 3-5 the reading is not clear

Our response: The size of the tables are increased to improve the clarity of the numbers

Round 2

Reviewer 1 Report

1. In materials and method, 80 g ml-1 is total not possible. Please explain.

2. Please explain how the treated and control plants were separated, the single word barrier, does not explain all. The major concern is it is a field experiment. 

3. As per the materials and method, surface irrigation was provided, with soil treatment, how sure two plots will not interact.

4. English need to be improved. What is this statement means: Weed were mechanically and chemically (atrazine 4 L/ha) were applied

5. The methodology of sugar estimation is not reproducible, is this method from extraction, elution and quantification is standadized in your lab,

6. The citation of ref 51 needs to be verified.

Author Response

  1. In materials and method, 80 g ml-1 is total not possible. Please explain.

Our response: Thanks for valuable observation, it is 80 μg/ml

  1. Please explain how the treated and control plants were separated, the single word barrier, does not explain all. The major concern is it is a field experiment. 

Our response: We left a distance of two meters between different treatments, moreover we placed a strong plastic barrier of one meter deep in the soil and half a meter high above the soil surface.

  1. As per the materials and method, surface irrigation was provided, with soil treatment, how sure two plots will not interact.

Our response: We mean by "surface irrigation" that the watering was manually done from the highest point of each plot, so the plots couldn't interact with each other, especially since there were plastic barriers between them, and they were far apart.

  1. English need to be improved. What is this statement means: Weed were mechanically and chemically (atrazine 4 L/ha) were applied

Our response: The language has been reviewed by an expert and what has been changed is highlited by green colour.

The sentence has been changed to: “To prevent weeds competing and interfering with different treatments, weed were mechanically and chemically (treatment with atrazine 4 L/ha) removed”.

  1. The methodology of sugar estimation is not reproducible, is this method from extraction, elution and quantification is standardized in your lab.

Our response: More details were added to methodology, this method already documented in many labs including our lab. 

- Matros, Andrea, Darin Peshev, Manuela Peukert, Hans‐Peter Mock, and Wim Van den Ende. "Sugars as hydroxyl radical scavengers: proof‐of‐concept by studying the fate of sucralose in Arabidopsis." The Plant Journal 82, no. 5 (2015): 822-839.

- Ufarté, L., Bozonnet, S., Laville, E., Cecchini, D. A., Pizzut-Serin, S., Jacquiod, S., ... & Potocki Veronese, G. (2016). Functional metagenomics: construction and high-throughput screening of fosmid libraries for discovery of novel carbohydrate-active enzymes. In Microbial Environmental Genomics (MEG) (pp. 257-271). Humana Press, New York, NY.

- Srisimarat, Wiraya, Areeya Powviriyakul, Jarunee Kaulpiboon, Kuakarun Krusong, Wolfgang Zimmermann, and Piamsook Pongsawasdi. "A novel amylomaltase from Corynebacterium glutamicum and analysis of the large-ring cyclodextrin products." Journal of inclusion phenomena and macrocyclic chemistry 70, no. 3 (2011): 369-375.

-AbdElgawad, Hamada, Darin Peshev, Gaurav Zinta, Wim Van den Ende, Ivan A. Janssens, and Han Asard. "Climate extreme effects on the chemical composition of temperate grassland species under ambient and elevated CO2: a comparison of fructan and non-fructan accumulators." PLoS One 9, no. 3 (2014): e92044.

  1. The citation of ref 51 needs to be verified.

Our response: El-Henawy, A.; El-Sheikh, I.; Hassan, A.; Madein, A.; El-Sheikh, A.; El-Yamany, A.; Radwan, A., Mohamed, F.; Khamees, M.; Ramadan, M.; Abdelhamid, M.; Khaled, H.; El-Faramawy, Ayoub, H.Y.; Youssef, S.; Faizy, S.E.D. Response of cultivated broccoli and red cabbage crops to mineral, organic and nano-fertilizers. Env. Biodiv. Soil Security 2018, 221-23. doi.10.21608/JENVBS.2019.6797.1046 has taken the place of Reference No. 51.

Reviewer 2 Report

 Accept in present form

Author Response

We would like to thank the first reviewer for accepting the manuscript in its previous version.

Round 3

Reviewer 1 Report

-